# Long-Term Ventilation in Children with Medical Complexity: A Challenging Issue

**DOI:** 10.3390/children9111700

**Published:** 2022-11-05

**Authors:** Valeria Caldarelli, Federica Porcaro, Paola Di Filippo, Marina Attanasi, Valentina Fainardi, Marcella Gallucci, Angelo Mazza, Nicola Ullmann, Stefania La Grutta

**Affiliations:** 1Department of Mother and Child, Azienda USL-IRCCS di Reggio Emilia, 42123 Reggio Emilia, Italy; 2Pediatric Pulmonology & Respiratory Intermediate Care Unit, Sleep and Long-Term Ventilation Unit, Academic Department of Pediatrics, Bambino Gesù Children’s Hospital, IRCCS, 00146 Rome, Italy; 3Department of Pediatrics, SS Annunziata Hospital, University of Chieti, 66100 Chieti, Italy; 4Cystic Fibrosis Unit, Department of Paediatrics, Parma University Hospital, 43126 Parma, Italy; 5Department of Pediatrics, S. Orsola-Malpighi Hospital, University of Bologna, 40126 Bologna, Italy; 6Department of Pediatrics, ASST Papa Giovanni XXIII, 24127 Bergamo, Italy; 7Institute of Traslational Pharmacology IFT, National Research Council, 90146 Palermo, Italy

**Keywords:** children with medical complexity, continuous positive airway pressure, long-term invasive mechanical ventilation, sleep apnea syndrome, sleep-disordered breathing, palliative care, ethical issues

## Abstract

Children with medical complexity (CMCs) represent a subgroup of children who may have congenital or acquired multisystemic disease. CMCs are frequently predisposed to respiratory problems and often require long-term mechanical ventilation (LTMV). The indications for LTMV in CMCs are increasing, but gathering evidence about indications, titration, and monitoring is currently the most difficult challenge due to the absence of validated data. The aim of this review was to examine the clinical indications and ethical considerations for the initiation, continuation, or withdrawal of LTMV among CMCs. The decision to initiate long-term ventilation should always be based on clinical and ethical considerations and should be shared with the parents.

## 1. Introduction

Children with medical complexity (CMCs) is a term first used in 2011 [1] to describe a subgroup of children who may have congenital or acquired multisystemic disease. CMCs include children with four principal characteristics [1,2]: chronic disease, serious limitations in functionality requiring technological assistance for usual activities, significant healthcare needs, and substantial utilization of healthcare resources. CMCs are a fast-growing subpopulation and represent 0.4% to 0.7% of all US children [3]. In light of this, Imperial College London carried out the Models of Child Health Appraised (MOCHA) project to analyze the typical approaches in 30 European Union (EU) and European Economic Area (EEA) countries in planning the care of CMCs [4].

Although the prevalence of CMCs for each country is not currently known, it is known that 4.2 per 100,000 Italian children are assisted by technology because they require long-term mechanical ventilation (LTMV) [5]. LTMV is ventilation delivered for a minimum of six hours per day for more than three weeks. CMCs are frequently predisposed to respiratory problems (aspiration pneumonia, sleep-disordered breathing, impaired cough, recurrent respiratory infections, and respiratory failure), some of which may require LTMV [6,7]. In addition, we must not forget that CMCs on LTMV are at risk of several complications (poor growth, developmental delay, tracheostomy plugging, accidental decannulation, pulmonary hypertension, and sepsis), which may worsen existing conditions and even induce cardiorespiratory arrest. 

Accordingly, the decision to begin or forgo LTMV for CMCs can be difficult and impactful. Clinicians should help families to understand the different options and their consequences [8]. Families may reject mechanical ventilation because of its ineffectiveness or the difficulty in its management, even in non-terminal cases [8]. The decision-making process regarding the initiation, continuation, or withdrawal of LTMV should be (1) carried out in an interdisciplinary and unhurried manner, (2) based on the prognosis of the disease, (3) shared with the family and the patient (when possible), and (4) started as soon as chronic respiratory failure is anticipated or diagnosed [8]. 

In any case, the aim of LTMV in a CMC is to improve the child’s quality of life (realizing better control of hypercapnia and hypoxemia, reducing respiratory fatigue, and promoting growth and development), not just to prolong life, enabling the patient to stay at home and avoid excessive medicalization [9,10,11]. However, delivering such care is not without burdens and risks, and providers should help caregivers to be aware of the complex care coordination and significant resource utilization that LTMV requires [12].

Although the use of LTMV in the pediatric field is ever increasing, the gathering of evidence about the indications, titration, and monitoring of LTMV in CMCs is currently very challenging due to the absence of validated data. The aim of this review was to examine the clinical indications and ethical considerations for the initiation, continuation, or withdrawal of LTMV in CMCs. The development of studies on LTMV in integral care, from a clinical and ethical perspective, will define its role in the healthcare of CMCs [13].

## 2. Sleep-Disordered Breathing

Sleep-disordered breathing (SDB) represents a real concern for CMCs, as they often have alterations in neuromuscular tone, dysfunctions of the central nervous system (CNS), and abnormalities in craniofacial structures [14,15]. The term SDB encompasses a spectrum of ventilatory disorders, such as primary snoring, obstructive sleep apnea syndrome (OSAS), central sleep apnea (CSA), and hypoventilation syndrome (HS) [16,17], which are often associated with multisystemic complications.

Primary snoring is characterized by habitual snoring (more than 3 nights per week) without apneas or hypopneas [18]. The incidence of snoring and OSAS in the pediatric population is around 2%, making it the most common subtype of SDB [1,18]. However, OSAS can be up to ten times more prevalent in a subset of CMCs [1,19] with anatomical and/or functional risk factors (adeno-tonsillar hypertrophy, crowding of the midface structures, reduced dimension of the upper airways with relative macroglossia, and generalized muscular hypotonia), increasing the upper airway collapsibility and, accordingly, predisposing them to SDB [20,21,22].

CSA occurs in about 1–5% of healthy children but is more common in CMCs, who often have bulbar dysfunction with consequent impaired breathing control [19]. The result of this condition is a failure of respiratory drive during sleep, resulting in a drop in arterial oxygen levels, frequent nocturnal awakenings, arousals, and sleep fragmentation [19].

HS is usually determined by a mismatch between the mechanical demand and ventilatory response. The two mechanisms—that can be present individually or in association—contribute to levels of alveolar ventilation that do not allow the maintenance of normal gas exchange, possibly resulting in hypoxemia and hypercapnia. Central congenital hypoventilation syndrome, neuromuscular disorders, or thoracic cage disorders are most associated with these forms of SDB. However, CMCs with chronic ventilatory disorders are prone to nocturnal hypoventilation, predictive of the development of respiratory failure in the daytime. This last condition is defined as a transcutaneous and/or end-tidal carbon dioxide recording of more than 50 mmHg for more than the 25% of the total sleep time in pediatric age [22].

Based on the described disorders, LTMV seems to be necessary to safely manage CMCs affected by SDB at home, as reported in Table 1.

## 3. Indications of Long-Term Non-Invasive Ventilation in CMC

Long-term non-invasive ventilation (LTNIV) and its development have reduced the need for invasive ventilation [13,23,24,25]. LTNIV includes two modes of ventilation: continuous positive airway pressure (CPAP) and non-invasive ventilation (NIV).

CPAP and NIV are widely used in children, but why and when these treatments should begin are not well defined for pediatric-age patients. A decrease in central drive and/or increases in respiratory load and muscular weakness all contribute to a respiratory imbalance leading to respiratory failure [13,25] that could possibly benefit from ventilatory support.

Based on the rate at which the decline in respiratory function occurs, CPAP and NIV could be required in acute, subacute, or chronic settings. For the last two, the decision to start ventilatory support can be prompted by different respiratory parameters, such as SpO_2_, PtcCO_2_, and/or AHI [13,25].

Although the criteria leading to CPAP or NIV initiation in pediatrics are often adapted from those used in adults, the presence of various clinical symptoms (i.e., paradoxical breathing, poor sleep quality, morning headaches, and waking with breathlessness) suggests the need to start LTNIV [26]. In addition, abnormal findings in arterial blood gas and sleep studies (polysomnography or nocturnal pulse oximetry and capnography) support the decision to initiate ventilation and help clinicians choose the most appropriate mode of ventilation [27,28].

In particular, the updated respiratory scoring rules and the definitions of apnea or hypopnea events and periodic breathing for pediatric-age patients (Table 2) guide clinicians in deciding when to start the ventilatory support (Table 2) [29].

To date, the only absolute criterion requiring NIV initiation remains hypercapnia. However, American recommendations and guidelines providing guidance on the best time to start LTNIV are already available for some conditions previously described and reported in Table 3 [29,30,31].

## 4. Indication of Long-Term Invasive Mechanical Ventilation in CMCs

Long-term invasive mechanical ventilation (LTIMV) is delivered using a tracheostomy interface. An acute respiratory complication is a frequent, severe development in CMC patients, and invasive mechanical ventilation represents the standard treatment where initial management—with oxygen supplementation, physiotherapy, cough assistance, or antibacterial drugs—is insufficient to stabilize the patient [32]. In addition, tracheostomy in CMCs is usually considered in a sub-acute or chronic setting, either when failure to wean from mechanical ventilation (MV) via an endotracheal tube occurs or when MV will probably be used for prolonged periods and non-invasive ventilation appears inadequate [26,32].

Indeed, an Italian survey including 535 children from 57 centers confirmed that failure to wean from MV was the primary indication for invasive ventilation through tracheostomy and that a younger age, a need for near continuous ventilatory assistance, the presence of cerebral palsy (CP), excessive airway secretions, and impaired swallowing helped the clinicians to make the decision to use LTIMV [5].

Among CMCs with severe pulmonary diseases or/and acute exacerbations, invasive ventilation may improve gas exchange and respiratory distress and may sometimes be lifesaving, although a careful assessment of the risks should be performed in each child. The advantages of this approach are improvements in the removal of secretions, protection of the airways, and increased comfort for both the patients and caregivers in general management. Nevertheless, it is generally the last respiratory assistance choice for CMCs [33].

The ATS Guidelines on Pediatric Chronic Home Invasive Ventilation recognize that children incapable of maintaining normocapnia and/or adequate oxygen saturation because of reduced ventilation need LTIMV [33]. 

Patients with chronic respiratory insufficiency may present with continuous or sleep-related inadequate ventilation [34].

Most of the literature reports the use of LTIMV in three main disease categories: neuromuscular disorders (NMDs), CSA, and chronic bronchopulmonary diseases such as bronchopulmonary dysplasia (BPD), cystic fibrosis (CF), and genetic disorders. Indeed, a Canadian pediatric home ventilation program reported NMDs as the first indication for LTIMV (37% of subjects), followed by CSA and bronchopulmonary respiratory diseases (31% each) [35,36].

An Italian cohort study published in 2011 included 162 pediatric patients receiving LTIMV as a result of NMD (45%), CP (21%), and CSA (15%) [5]. In a very recent Italian study by Pavone et al., children on LTIMV were affected by NMDs (30.6%), upper respiratory airway diseases (24.8%), and CNS diseases (22.7%) [32].

The improvement in CO_2_ levels, daytime neurological functions, symptoms related to hypercapnia, and life expectancy in all of the above reported conditions make clear the indication for LTIMV (Table 4), which should be carefully assessed on the basis of its risk/benefit ratios in these groups of patients [32].

## 5. Palliative Care and Ethical Issues in Long-Term Ventilation in CMCs

The World Health Organization (WHO) defines palliative care (PC) as an approach aiming to improve the quality of life (QoL) of patients with life-threatening illnesses and their families. Improved QoL is achieved by the prevention and relief of pain through the early identification, assessment, and treatment of possible problems (physical, psychosocial, and spiritual) [33].

In children with CMCs, the main goals of PC (Figure 1) are achieving the best possible QoL, alleviating physical and psychosocial distress, understanding the needs of children and their families, and providing adequate end-of-life care.

When dealing with children with CMCs, the role of PC should be integrated a priori into an overall advanced care plan that can be adapted according to the child’s condition. If a patient with LTNIV deteriorates, switching from LTNIV to LTIMV often necessitates a tracheostomy, with consequent changes in life expectancy and the patient’s and family’s QoL. This is, therefore, a crucial moment in the treatment of CMC patients, at which decisions about respiratory assistance must be thoroughly discussed and shared with the family and, when possible, the patient. Honest communication and complete information about the treatment and prognosis are the basics regarding PC and end-of-life care. In the case of failure to share the decision about the transition to tracheostomy to allow LTIMV, LTNIV can be continued as part of PC [33].

Children with insufficient mental and/or physical functions can represent difficulty in engagement for clinicians in the end-of-life process because medical decisions are delegated to the parent or caregiver. In more cognitively able children, information on clinical status must be provided and age-appropriate methods, such as books and visual aids, must be employed to enhance their expressive communication skills [34]. However, some patients with life-threatening diseases may prefer not to be consulted [34]. For this reason, the child should be encouraged to express his or her opinion about their desired level of involvement in the decision-making process.

The support of a multidisciplinary team is essential for properly moderating discussion about PC and end-of-life decisions with both the patient and the caregivers. The decision to switch from LTNIV to LTIMV and the transition to tracheostomy in a CMC is very challenging and associated with a great burden on the child and family.

The American Academy of Pediatrics (AAP) has recently issued new recommendations concerning ethical considerations as well as available scientific evidence to guide clinicians facing the shared decision-making process with the families of CMCs [34]. In the discussion, physicians must take into account emotions; cultural, spiritual, and religious beliefs; and the past experience of the caregivers, and must make the most medically appropriate decisions to promote the best choice for the child.

The cornerstones of shared good decision making are collaboration, effective communication, and respect for ethical principles. The last include respect for life and always looking for the best solution for the child, which means preventing prolonged suffering, either when death is imminent or when treatment would not provide benefit to the patient, making life intolerable because of pain. 

Since the regulations, guidelines, and approaches may vary between countries and hospitals, the hospital ethics committee can assist clinicians, patients, and their caregivers with the end-of-life decision making. Training in the ethical aspects of treatment as well as palliative care is crucial for healthcare providers to guarantee a targeted approach in every condition [35].

More research is needed to reach a consensus about the shared decision-making process, including the identification of the best outcomes that can objectively support the validity of the process, such as child and family satisfaction [36,37].

## 6. Conclusions

The increased survival of a subgroup of children with complex medical conditions requires long-term ventilation. The use of non-invasive ventilatory support is increasing worldwide, permitting easy and direct management of the child at home, increasing the quality of life for the families affected. Nevertheless, invasive ventilation delivered through tracheostomy may be a necessary choice in some circumstances in order to improve a patient’s survival. The decision to start long-term ventilation should always be based on clinical and ethical considerations and should be shared with the parents, the patient (when possible), and all the figures directly involved in the home management of the child with medical complexity.

## Figures and Tables

**Figure 1 children-09-01700-f001:**
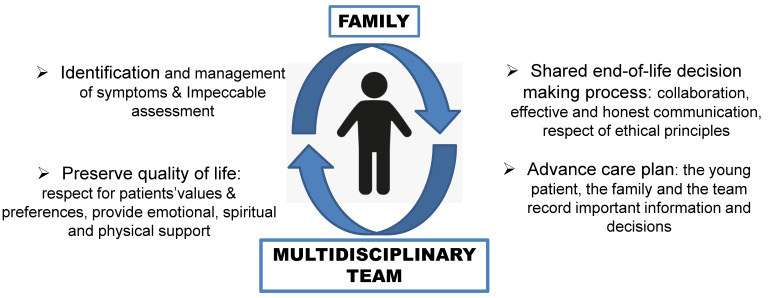
Visual summary of palliative care.

**Table 1 children-09-01700-t001:** Sleep-disordered breathing in CMCs.

Sleep-Disordered Breathing	Definition	Physiopathology
OSAS	Recurrent partial or complete upper-airway obstruction (hypopneas, or obstructive or mixed apneas) with disruption of normal oxygenation, ventilation, and sleep patterns.	Structural and functional factors that increase upper-airway collapsibility. Frequent in NMD and craniofacial abnormalities.
CSA	Absence of central respiratory drive due to a transient reduction by the pontomedullary respiratory rhythm generator.	Bulbar dysfunction and a lack of drive to breathe during sleep result in a drop in arterial oxygen levels, frequent nocturnal awakenings, arousals, and sleep fragmentation.
HS	Persistent low-tidal-volume breathing or bradypnea resulting in hypercarbia and hypoxemia. Nocturnal hypoventilation: transcutaneous and/or end-tidal carbon dioxide recording of >50 mmHg for >25% of the total sleep time.	An increased mechanical load of breathing and decreased ventilatory drive/response contribute to levels of alveolar ventilation inadequate for maintaining normal gas exchange. Central congenital hypoventilation syndrome, neuromuscular disorders, or thoracic cage disorders are most associated with this SDB.

OSAS = obstructive sleep apnea syndrome; CSA = central sleep apnea; HS = hypoventilation syndrome.

**Table 2 children-09-01700-t002:** American Academy of Sleep Medicine (AASM): Updated pediatric scoring rules [18].

Definition of Respiratory Events	
Apnea	A drop in the peak signal excursion by ≥90% from the pre-event baseline. The duration of the ≥90% drop lasts at least the duration of two breaths and is associated with the presence of respiratory effort throughout the entire period of absent airflow.
Central Apnea	Apnea with an absence of inspiratory effort throughout the duration of the event, and at least one of the following conditions is met:
	The event lasts 20 s or longer. The event lasts at least the duration of two breaths during baseline breathing and is associated with an arousal or ≥3% oxygen desaturation. For infants younger than 1 year of age, the event lasts at least the duration of two breaths during baseline breathing.
Mixed apnea	Apnea criteria for at least the duration of two breaths during baseline breathing, associated with an absence of respiratory effort during one portion of the event and the presence of inspiratory effort in another portion.
Hypopnea	A 30% drop in flow with either ≥3% oxygen desaturation or arousal lasting at least 2 breaths during baseline breathing.
Hypoventilation	>25% of the total sleep time is spent with a PCO_2_ > 50 mmHg.
Periodic breathing	≥3 episodes of central apnea lasting >3 s separated by no more than 20 s of normal breathing.

**Table 3 children-09-01700-t003:** Indications for long-term non-invasive ventilation initiation in obstructive sleep apnea syndrome, spinal muscular atrophy and Duchenne muscular dystrophy.

Disease	Indications
OSAS[^30^]	Severe, RDI > 10. Hypercapnia.
SMA[^30^]	Daytime hypercapnia (i.e., diurnal PCO_2_ > 45 mmHg). Sleep hypoventilation. Obstructive sleep apnea. Paradoxical breathing and chest wall deformity. Recurrent chest infections requiring hospital admission (>3 per year). Failure to thrive.
	In patients already using nocturnal NIV, daytime NIV is indicated for:
	The self-extension of nocturnal ventilation into waking hours. Abnormal deglutition due to dyspnea, which is relieved by NIV. An inability to speak a full sentence without breathlessness. Symptoms of hypoventilation with baseline SpO_2_ < 95% and/or PCO_2_ > 45 mmHg while awake.
DMD[^30^]	Early signs or symptoms of sleep hypoventilation. Abnormal sleep study results. An FVC < 50% predicted. An MIP < 60 cm H_2_O. An awake baseline SpO_2_ < 95% or pCO_2_ > 45 mmHg.
	The daytime NIV should be added when, despite nocturnal ventilation, the daytime SpO_2_ is < 95%, pCO_2_ > 45 mm Hg, or symptoms of awake dyspnea are present. In addition to the above criteria, NIV is indicated when:
	The end-tidal or transcutaneous carbon dioxide (CO_2_) is >50 mm Hg for ≥2% of the sleep time. A sleep-related increase in end-tidal or transcutaneous CO_2_ of 10 mm Hg above the awake baseline for ≥2% of sleep time is observed. The SpO_2_ is ≤88% for ≥2% of sleep time or for at least 5 min continuously. The apnea–hypopnea index is ≥5 events per hour.

RDI = respiratory disturbance index; OSAS = obstructive sleep apnea syndrome; SMA = spinal muscular atrophy; PCO_2_ = partial pressure of carbon dioxide; NIV = non-invasive ventilation; SpO_2_ = oxygen saturation; DMD = Duchenne muscular dystrophy; FVC = forced vital capacity; MIP = maximal inspiratory pressure.

**Table 4 children-09-01700-t004:** Indications for long-term invasive mechanical ventilation initiation in children with medical complexity.

Long-Term Invasive Mechanical Ventilation			
	Central Nervous System Diseases	Musculoskeletal Diseases	Respiratory Diseases
**Acute** **Indications**	Acute exacerbations	Chest wall deformities Neuromuscular diseases	Restrictive, obstructive, or mixed disorders
**Chronic** **Indications**	Spinal cord injuries Congenital central hypoventilation syndrome	Chest wall deformities Neuromuscular diseases	Cystic fibrosis Bronchopulmonary dysplasia Severe tracheobronchomalacia

## Data Availability

No new data were created or analyzed in this study.

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
