# Peer review of "Long-Term Ventilation in Children with Medical Complexity: A Challenging Issue"

_children, 2022, doi:10.3390/children9111700_

Round 1

Reviewer 1 Report

The authors of this review article aim examine the clinical indications and the ethical considerations for initiation, continuation or withdrawn of  mechanical ventilation (MV) among children with medical complexity. In general, this article is provide important information about the available information to manage MV in pediatrics. The article needs extensive editing of English and style. The authors should consider to revise the outline to address the need of long term MV as follow: 

1. Introduction

2. Central Nervous System Diseases

3. Muscoloskeletal diseases

4. Respiratory Diseases

5. Ethic considerations

6. Conclusion

Figures and tables needs to be improved, void the use of different types of bullets points and improve the quality of the figures.

Avoid the use of "Technology Dependent children", insted use the term: "Children assisted to Technology. 

Remove appendix section if not used. 

Most of the articles discussed and presented in this review are from Italy. Your review will get better citation index if you discuss major paper from key countries and contrast them with Italy data. Otherwise may change the title to: 

Long term ventilation in ITALIAN children with medical complexity: a challenging issue

Also, I will consider a better title like:

Long term ventilation: A challenge in children with medical complexity

Reference #34 is used three times from 160-165.

Figure 1 should be reviewed and resubmitted after fixing several typos. Consider a better image to provide a more impacting figure. 

Author Response

Dear reviewer,

we are performing a large enghlish revision of the manuscript but we are convinced that the organization of the article not by patologies but for comcepts is clear enogh. Here you can find a major revision of tables and manuscript.

Reviewer 2 Report

Very nice article highlighting use of mechanical ventilation (either invasive or non-invasive) in children with medical complexities including citation of recent references on this important yet complex issue. I would have the article read thoroughly by an English editor to improve grammar, formality, and wording (including lines 47-48, 54-55, and multiple other areas throughout the paper). Would also update the formatting of tables 2 and 3 as different style bullets are used and the text being centered makes it difficult to read. Also, unclear if words underlined in Figure 1 text is intentional but would have the authors double check this.

Overall, this article is a nice review which likely warrants acceptance once grammar/style issues are resolved.

Author Response

(The authors gave the same response as above.)

Round 2

Reviewer 1 Report

Dear authors, at this point, all my recommendations were addressed. Minor spell check is still required.